# Electrochemistry-assisted selective butadiene hydrogenation with water

Yong-Qing Yan[1,4], Ya Chen[1,4], Zhao Wang [ID][1,2] ✉, Li-Hua Chen[1], Hao-Lin Tang[1,2] & Bao-Lian Su [ID][1,3] ✉

Alkene feedstocks are used to produce polymers with a market expected to reach 128.4 million metric tons by 2027. Butadiene is one of the impurities poisoning alkene polymerization catalysts and is usually removed by thermocatalytic selective hydrogenation. Excessive use of $H_2$, poor alkene selectivity and high operating temperature (e.g. up to 350 °C) remain the most significant drawbacks of the thermocatalytic process, calling for innovative alternatives. Here we report a room-temperature (25-30 °C) electrochemistry-assisted selective hydrogenation process in a gas-fed fixed bed reactor, using water as the hydrogen source. Using a palladium membrane as the catalyst, this process offers a robust catalytic performance for selective butadiene hydrogenation, with alkene selectivity staying around 92% at a butadiene conversion above 97% for over 360 h of time on stream. The overall energy consumption of this process is 0.003 Wh/mL$_{butadiene}$, which is thousands of times lower than that of the thermocatalytic route. This study proposes an alternative electrochemical technology for industrial hydrogenation without the need for elevated temperature and hydrogen gas.

Light alkenes, especially propene, and butenes, are widely used as raw materials for the industrial synthesis of polymers[1]. They are mainly obtained by petroleum alkanes cracking reaction, which produces small amounts of alkynes and alkadienes as impurities (0.3%–8%). As these impurities can severely poison alkene polymerization catalysts (i.e., Ziegler-Natta catalysts[2]), their elimination to a concentration below 5 ppm is vital[3]. Currently, these impurities are removed by thermocatalytic hydrogenation reaction, which selectively converts alkynes and alkadienes to alkenes, and avoids alkanes and polymers formation. The high cost, low alkenes selectivity at high alkynes/alkadienes conversion and quick deactivation of the commonly-used palladium catalyst are prompting researchers to focus on exploring alternative catalysts, such as supported non-noble metal catalysts[4,5] or supported single-atom catalysts (SACs)[6–10]. Unfortunately, the explored catalysts need an elevated operating temperature and an excess of hydrogen gas, leading to serious safety issues[11]. Hence, the

development of a new process that offers high alkenes selectivity, catalytic stability, low energy consumption, and high safety is a pressing and challenging demand.

Utilization of water as an alternative hydrogen source for industrial selective hydrogenation is a highly desired perspective. The electrocatalytic hydrogenation of organics with hydrogen atoms (labeled as $H_a$) that are in-situ generated on the surface of the electrode by water electrolysis has been explored either in the same reactor with water electrolysis[12,13], or in two separate reactors[14–17]. However, all these reactions are carried out in liquid-phase and in a batch system, which creates unavoidable problems that impede industrial implementation, such as separation of the target products from reactants, low reaction efficiency at the liquid/solid interface, low reproducibility, and poor selectivity.

Here, we report a room temperature (25-30 °C) electrochemistry-assisted selective hydrogenation process, using water as a hydrogen

[1]State Key Laboratory of Advanced Technology for Materials Synthesis and Processing, School of Materials Science and Engineering, Wuhan University of Technology, 122, Luoshi Road, Wuhan 430070, China. [2]Foshan Xianhu Laboratory of the Advanced Energy Science and Technology, Guangdong Laboratory, Xianhu Hydrogen Valley, Foshan 528200, P. R. China. [3]Laboratory of Inorganic Materials Chemistry (CMI), University of Namur, B-5000 Namur, Belgium. [4]These authors contributed equally: Yong-Qing Yan, Ya Chen. ✉e-mail: zhao.wang@whut.edu.cn; bao-lian.su@unamur.be

source (Fig. 1a and Supplementary Fig. 1). This process offers an impressive hydrogenation performance, with ~92% of alkenes selectivity at above 97% of butadiene conversion at room temperature and with long-term catalytic stability. Such a process shows a very low energy consumption of 0.003 Wh/mL$_{butadiene}$, which is thousands of times lower than that of the current state-of-the-art thermocatalytic route (Supplementary Table. 1). Furthermore, safety hazards can be avoided because no hydrogen gas is used and generated in this method. The present work reports a highly selective, efficient, and sustainable hydrogenation process for the industrial elimination of alkynes and alkadienes present as impurities in alkenes polymerization.

## Results

### Electrochemistry-assisted selective hydrogenation

Figure 1b shows the schematic representation of the gas-feed fixed bed reactor, which is the core device designed for electrochemistry-assisted selective hydrogenation, using water as source of activated hydrogen atoms. It contains a liquid-phase cell for flowing electrolyte, a gas-phase cell for feeding gas reactants, and a compact catalytic membrane sandwiched between two cells. The electrochemistry-assisted hydrogenation process based on this novel reactor design is schematically shown in Fig. 1c. Specifically, water electrolysis occurs at the surface of catalytic membrane on the side of liquid-phase cell under loaded current density ($D_i$), offering a specific concentration ($C_0$) of adsorbed hydrogen atoms ($H_a$). The formed $H_a$ diffuse through the catalytic membrane to the opposite side and immediately react with the unsaturated hydrocarbons (e.g., 6000 ppm of butadiene in propene/helium gas mixture flux) on the surface of catalytic membrane in gas-phase cell. Three key capabilities are required for the membrane: (i) activity for hydrogen evolution reaction in water electrolysis; (ii) selectivity for $H_a$ penetration; (iii) activity for catalytic hydrogenation reaction. For the proof of concept, commercially available palladium membrane (thickness ~25 μm), containing a poly-

crystalline structure and a rough surface morphology (Supplementary Fig. 2–3) was used, owing to the well-known high catalytic activity of palladium towards hydrogenation and the hydrogen atom sieving property of palladium membrane.

Figure 2 presents the performance of the designed electrochemistry-assisted selective hydrogenation process with water, using selective butadiene semi-hydrogenation to butenes as a probe reaction. The catalytic behavior as a function of the loaded current density ($D_i$) in liquid cell is presented in Fig. 2a. Butadiene conversion gradually increases from 0 to ~10% at $D_i$ below −5 mA cm$^{-2}$, and it rapidly reaches above 90% at $D_i$ of 20 mA cm$^{-2}$. The catalytic selectivity to butenes maintains above 95% in the whole range of studied $D_i$. Similar catalytic behavior was also obtained by using 0.3% of acetylene in an excess of 3% propene with helium as balance gas (Supplementary Fig. 4). In addition, the catalytic activity of Pd was evaluated by the calculation of turnover frequency (TOF) in Fig. 2a. The TOF increases from 0.08 to 14.9 s$^{-1}$ upon increasing the $D_i$ from −0.05 to −20 mA cm$^{-2}$, with a critical TOF of 0.8 s$^{-1}$ at −5 mA cm$^{-2}$. Furthermore, the percentage of $H_a$ from water electrolysis that participates in the hydrogenation reaction was calculated and expressed as Faradaic efficiency (FE). The FE depicted in Fig. 2a has a sharp decrease from 100% to ~11.5% in the $D_i$ range of −0.05 to −5 mA cm$^{-2}$, followed by an increase to 52.3% at −15 mA cm$^{-2}$, and then a slight decrease to 47.2% at −20 mA cm$^{-2}$. In the reaction process, $D_i$ solely decides the formation of $H_a$ at the surface of Pd membrane in liquid cell. The formed $H_a$ could either penetrate through Pd membrane obeying Fick's first law for butadiene hydrogenation (i.e., contributing to the FE), or combine to form $H_2$ as the self-supported Pd membrane has to be assembled into the reactor for separating reactants gas and electrolyte liquid-phase. Detailed discussion on the $H_a$ penetration in Pd membrane under various $D_i$ is summarized in Supplementary Discussion 1. In detail, it was found that, at low $D_i$ (e.g., <0.15 mA cm$^{-2}$), no $H_2$ evolution was revealed by the cyclic voltammetry (CV) analysis (Supplementary Fig. 5), hydrogen atoms first prefer to laterally diffuse within the atomic layer of palladium lattice and then longitudinally permeate

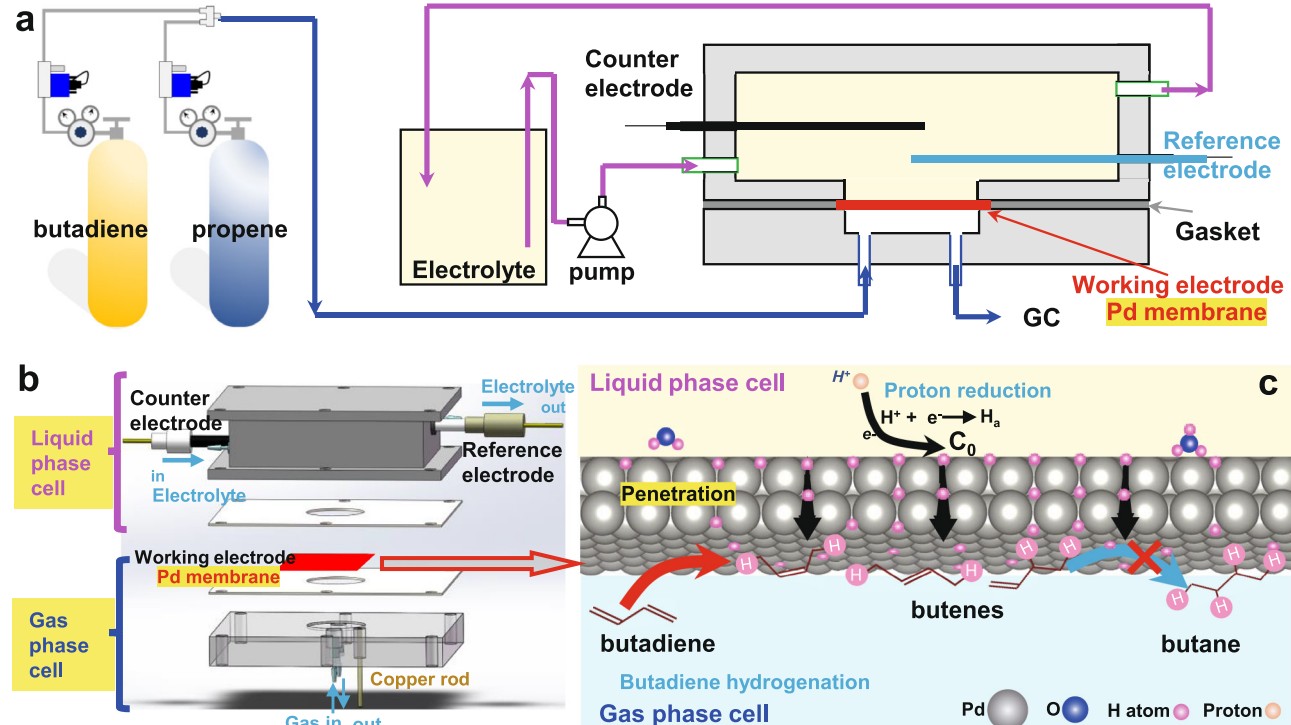

**Fig. 1 | Electrochemistry-assisted selective hydrogenation scheme. a** Scheme of the electrochemical-assisted selective butadiene hydrogenation system. **b** Construction of the gas feed fixed bed reactor. **c** The proposed reaction mechanism.

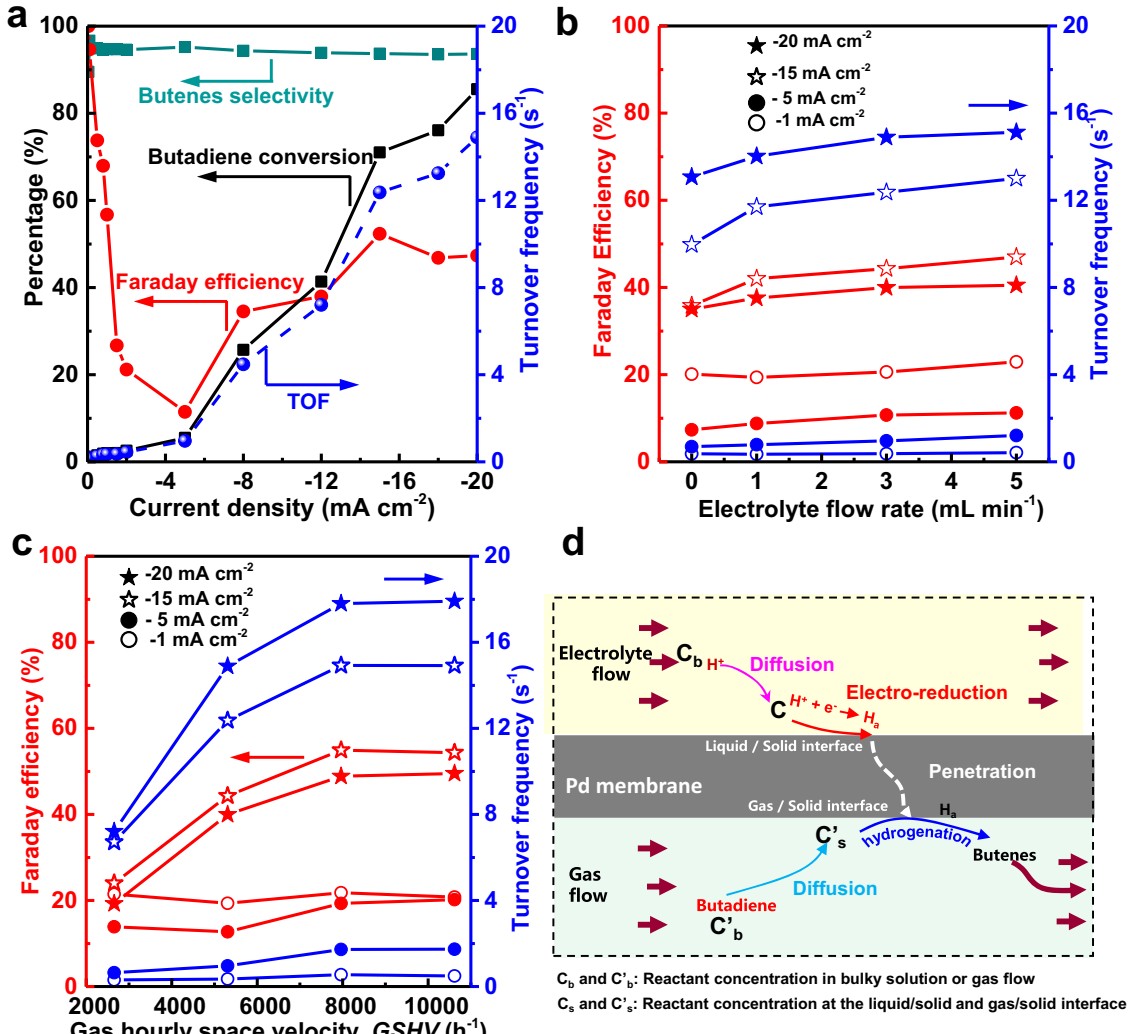

**Fig. 2 | Catalytic activity and selectivity to butenes of butadiene hydrogenation reaction. a** The effect of the current density for water electrolysis on the butadiene hydrogenation performances (0.6% of butadiene in helium, electrolyte flow rate = 3 mL min⁻¹ and the GHSV = 2654 h⁻¹. **b** The effect of the electrolyte flow rate on the butadiene hydrogenation performance at different current densities (0.6% butadiene in helium, GHSV = 2654 h⁻¹). **c** The effect of the GHSV on the butadiene hydrogenation performance at different current densities (0.6% butadiene in helium, electrolyte flow = 3 mL min⁻¹). **d** The process of electrochemistry-assisted hydrogenation reaction.

through the palladium membrane owing to its low energy barrier of diffusion (0.16 eV as shown in Supplementary Fig. 6). This offered a high utilization of $H_a$ (high FE at low Di). With the increase of $D_i$ offering a rapid $H_a$ formation and its accumulation on Pd surface, hydrogen atoms tend to directly penetrate palladium lattice and vertically pass through Pd membrane for catalytic reaction (Supplementary Discussion 1). Moreover, the increased $D_i$ could promote $H_2$ formation on the interface of Pd membrane to the electrolyte liquid-phase, and it further decreases the percentage of $H_a$ for butadiene hydrogenation. Specially, a phase transformation from Pd to $PdH_{0.75}$ was identified above $D_i$ −5mA cm⁻² (Supplementary Fig. 7a), it shows a swelling of palladium lattice from 3.89 Å of Pd to 4.02 Å of $PdH_{0.75}$. The interstitial hole in between octahedron void and tetrahedral void in Pd crystal increases from 0.217 Å to 0.315 Å (Supplementary Fig. 16c), which is larger than the diameter of hydrogen atom (0.25 Å), thus offering a continuous $H_a$ penetration. Therefore, the FE rapidly increase from 11.5% to 52.3% with $D_i$ rising from 5 mA cm⁻² to 15mAcm⁻². However, the higher $D_i$ value (e.g., >15 mA cm⁻²) largely promotes a strong $H_2$ evolution from $H_a$ combination[18], which is confirmed by the appearance of violent hydrogen bubbling over the Pd surface in liquid cell. Comparing with the previous exploration on the electrocatalytic hydrogenation reaction in liquid-phase (e.g., above

200 mA cm⁻²[15-17,19,20]), our process shows a ten times lower energy consumption, with above 90% of unsaturated hydrocarbon conversion at $D_i$ below 20 mA cm⁻². More importantly, the in-situ generation of active hydrogen atoms from water electrolysis, followed by longitudinal diffusion from liquid-phase side to gas-phase side of Pd membrane, plays a key role for obtaining such high catalytic hydrogenation performance. Supplementary Fig. 8 shows the catalytic preformance by using other hydrogen sources (e.g., premixing $H_2$ with reactants and inletting $H_2$ for $H_a$ penetration in Pd membrane), and it demonstrates that both premixing $H_2$ with butadiene (0.6% of butadiene/20% of $H_2$ mixture, balance with He) and using $H_2$ dissociation as the source of $H_a$ penetration show a rather low catalytic activities at room temperature (25-30 °C), with butadiene conversion below 10% and 1%, respectively. The advantage of our process can thus be attributed to the high efficiency of the active hydrogen atoms formation and the longitudinal diffusion for hydrogenation reaction, with detailed discussion included in Supplementary Discussion 2. The thickness of Pd membrane has a significant effect on the efficiency of the hydrogenation reaction. Thinner Pd membrane can decrease the distance of longitudinal diffusion and accelerate $H_a$ penetration through Pd membrane, thus enhancing the efficiency of hydrogenation reaction. However, the mechanical strength and compactness

of self-supported Pd membrane should also be taken into account. The catalytic performance of electrochemistry-assisted selective hydrogenation was further improved by optimizing the transportation of reactants (i.e., electrolyte in liquid-phase cell and the butadiene in gas-phase cell), as the heterogeneous reaction mainly occurs at the liquid/solid and gas/solid interface. Specifically, the catalytic activity (i.e., TOF) and the evolution of FE of $H_a$ utilization with the flow rate of electrolyte in the liquid-phase cell was analyzed in Fig. 2b. It shows that accelerating electrolyte flow brings an increase of catalytic activity (i.e., TOF), especially at a current density above 15 mA cm$^{-2}$. For instance, the TOF increases from around 10 s$^{-1}$ to above 13 s$^{-1}$ with rising electrolyte flow from 0 to 3 mL min$^{-1}$. It remains constant at flow rate above 3 mL min$^{-1}$, indicating that electrolyte transportation from bulky solution to the surface of palladium membrane is not the rate-determining step in the whole catalytic hydrogenation reaction[21–23]. Besides that, purging butadiene reactant with large flux (expressed as gas hourly space velocity, GHSV) was also found to increase TOF to above 17 s$^{-1}$ (Fig. 2c). This increase in TOF mainly happens at high $D_i$ and low GHSV range (<7964 h$^{-1}$) as the butadiene hydrogenation reaction was under butadiene transportation control at this stage. As for the $H_a$ utilization for butadiene hydrogenation (i.e., FE), both high flow rate of electrolyte in liquid-phase cell and butadiene reactant in gas-phase cell offer an improved FE, with a critical point at $D_i$ -15 mA cm$^{-2}$. DFT calculation in Supplementary Fig. 9 shows that the activation energy ($E_a$) for butadiene hydrogenation over Pd surface (i.e., Pd(111) and Pd(335)) is -0.8 eV, which is much lower than that of $H_a$ combination to $H_2$ (1.08 eV), suggesting that $H_a$ tends to react with butadiene, instead of forming $H_2$ in gas-phase cell, offering a high purity of reactants and products, without introducing exogenous impurities. The catalytic behavior, as shown in Fig. 2a–c, suggests that the catalytic performance of our electrochemistry-assisted hydrogenation route is significantly affected by three parts in the reactor, namely the mass transportation (i.e., liquid electrolyte and gas reactants) in liquid-phase cell and in gas-phase cell, the formation and penetration of hydrogen atoms over Pd membrane (Fig. 2d and Supplementary Fig. 1c). Optimization of these three parts could obtain enhanced catalytic performance. Therefore, the results presented in Fig. 2 indicate that our electrochemistry-assisted selective hydrogenation process highly boosts butadiene selective hydrogenation, offering high butadiene conversion (>90%) and high butenes selectivity (-95%) at room temperature (25-30 °C), with activated hydrogen atoms in-situ generated directly from water electrolysis at low current density (<15 mA cm$^{-2}$).

## Catalytic stability

The stability of catalytic performance is a critical parameter in determining its industrial applicability. Figure 3 compares the catalytic stability of butadiene selective hydrogenation in an excess of propene ($n_{butadiene}$:$n_{propene}$ -1:10) by both the electrochemistry-assisted selective hydrogenation process and the thermocatalytic route at room temperature (25-30 °C). The physico-chemical properties of commercial Pd/Al$_2$O$_3$ catalyst for the thermocatalytic reaction are presented in Supplementary Fig. 10, which shows a particle size of supported Pd around 10 nm. In the electrochemistry-assisted selective hydrogenation with water as hydrogen source, the initial butadiene conversion increases from ~90% to above 98.5% after 2 h of activation, with a selectivity to butenes at ~92%. An unprecedentedly stable catalytic activity and alkenes selectivity was observed for over 360 h of reaction. However, as for the currently-used industrial Pd/Al$_2$O$_3$ catalyst, it stays at 100% butadiene conversion in the first 50 h of reaction, and then rapidly decreases to below 10% after 80 h of hydrogenation reaction. In addition, our electrochemistry-assisted selective hydrogenation with water shows very high butenes selectivity of 92% at butadiene conversion above 97% and low propane formation of below 8% (Fig. 3b). In contrary, the commercial Pd/Al$_2$O$_3$ catalyst shows a poor selectivity to butenes, with high conversion of propene to propane in the first 50 h

of reaction. The catalytic selectivity to butenes gradually increases from 10% to above 98% with a significant drop in butadiene conversion from 100% to below 60%. Moreover, based on the voltage and the $D_i$ (Supplementary Fig. 11) used for the electrochemistry-assisted selective hydrogenation, the calculated energy consumed during stability test was 0.003 Wh/mL$_{butadiene}$, which is thousands of times lower than that of the current explored catalysts for such thermocatalytic selective hydrogenation reaction (Supplementary Table 1). In comparison with the thermocatalytic process[24] and recently reported electrocatalytic process[25,26], our electrochemistry-assisted process exhibits significant superiority in alkynes/alkadienes conversion efficiency, catalytic stability, alkenes selectivity, prevention of exogeneous impurities, reduction of energy consumption and improvement of safety (Fig. 3c). The Zielger-Natta catalysts for alkenes polymerization mainly consist of a mixture of an alkyl derivative of aluminum and titanium tetrachloride, which are easily hydrolyzed by the presence of water as impurity. Furthermore, the undesirable presence of excessive $H_2$ in alkenes could bring a termination of the alkenes polymerization[2], resulting in a low molecular weight of poly-alkenes product. It is therefore worth noting that the innovative electrochemistry-assisted process could avoid $H_2$ and $H_2O$ from butadiene reactants by using the special hydrogen atom sieving property of palladium membrane, thus offering a high purity of reactants and products (i.e., verification by the in-situ GC coupled with online mass-spectroscopy analysis on the outlet gas products in Supplementary Fig. 12).

## Catalytic mechanism

The superior catalytic performance of our electrochemistry-assisted hydrogenation process with water was further explored by the density functional theory (DFT) in Fig. 4. The adsorption energy ($E_{ads}$) of various guest species during butadiene hydrogenation reaction on Pd (111) surface was firstly calculated and presented in Fig. 4a, using the $E_{ads}$ of 2-butene as a baseline. It was found that, on Pd (111) surface, the $E_{ads}$ of various species follows: butadiene <hydrogen atom at the interstitial sites of face center cubic (H-fcc) ≈ 2-butene <cis-2-butene ≈ hydrogen atom at the interstitial sites of hexagonal closest-packed (H-hcp) <1-butene <$H_2$ < hydrogen atom at the atop sites of Pd atom (H-top) <butane. This indicates that butadiene has the priority to adsorb on the Pd surface in the reactants mixture. $H_2$ molecule has a weaker adsorption on Pd than that of butadiene and butenes (e.g., 2-butene, cis-2-butene and 1-butene). This explains the reason for which an excess of premixed $H_2$ is normally required for selective alkynes/alkadienes hydrogenation in the industrial thermocatalytic hydrogenation reaction, as the high $H_2$ partial pressure could promote $H_2$ adsorption and further its dissociation to adsorbed hydrogen atom[27,28]. Moreover, three types of adsorbed hydrogen atoms ($H_a$) were recognized on Pd surface (e.g., H-fcc, H-hcp and H-top). Among them, H-fcc is the most stable followed by H-hcp, and H-top. Therefore, with an increase of hydrogen atom number on Pd, the fcc site will be occupied in priority, followed by the hcp site, and finally the top site. The configuration of these hydrogen atoms is schematically represented in Fig. 4b. It shows that, with an increase of $H_a$ stability, $H_a$ tends to be close-packed into the interstitial hole of Pd atom, with the vertical distance between $H_a$ and surface Pd atom decreasing from 1.5405 to 0.7714 and 0.7614 Å for H-top, H-hcp and H-fcc, respectively.

The reaction procedure of selective hydrogenation of butadiene was further analyzed following the two classical models, i.e., the Eley-Rideal (E-R) model and the Langmuir-Hinshelwood (L-H) model by using 2-butene as a target product. Firstly, the E-R reaction model was simulated by setting hydrogen atom on the top site of Pd (H-top), butadiene molecule gradually approach to the surface of Pd and react with H-top. It was found that, with a decrease in the distance between butadiene and Pd surface (R$^{\#}$) from 4.498 Å to 3.406 Å, the $E_a$ of the calculated system has an increase of 0.12 eV. Importantly, at the R$^{\#}$ below 3.406 Å, DFT calculation shows that butadiene molecule automatically reacts

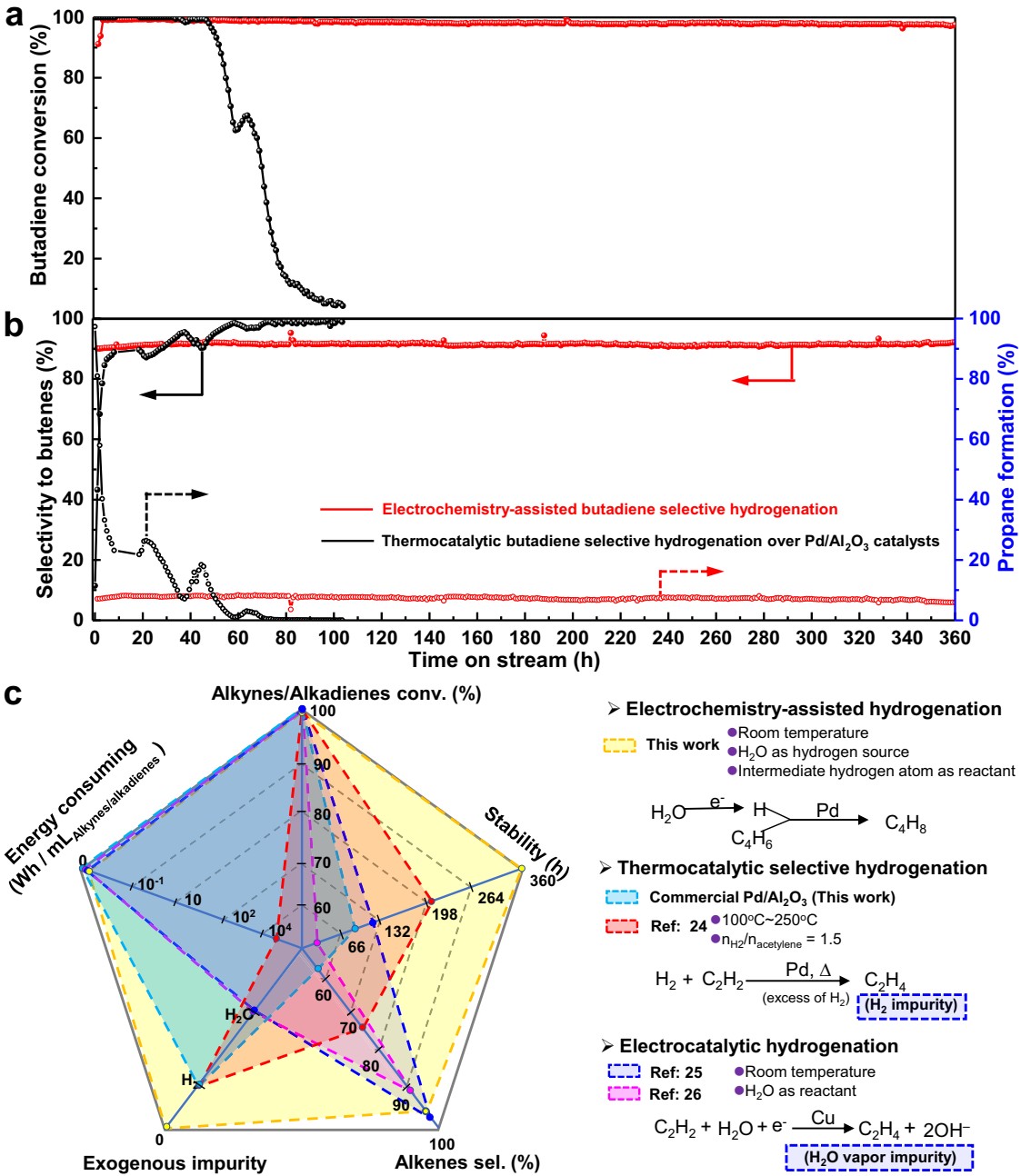

**Fig. 3 | Catalytic stability comparison between the electrochemical-assisted selective hydrogenation and the hydrogenation on commercial Pd/Al₂O₃ catalyst. a** Butadiene conversion and **b** selectivity to butenes and propane formation as a function of time on stream by electrochemistry-assisted selective hydrogenation strategy (current density = −15 mA cm⁻², electrolyte flow rate = 5 mL min⁻¹ and GHSV = 5308 h⁻¹, at 25-30 °C, with 0.6% butadiene and 6% propene in helium as reactants) and thermocatalytic selective hydrogenation reaction on commercial Pd/Al₂O₃ catalyst (25-30 °C, GHSV = 5308 h⁻¹, with a mixture of 0.6% butadiene, 6% propene and 20% hydrogen in helium). **c** Comparison of thermocatalytic, electrocatalytic and electrochemistry-assisted process in selective alkynes/alkadienes hydrogenation reaction.

with the adsorbed hydrogen atom, resulting in a decrease in the energy of the reaction system in Fig. 4c. The system was finally stabilized by releasing the semi-hydrogenation product of 2-butene. The energy evolution in the process of full hydrogenation of 2-butene to butane with H-top following the E-R model was also evaluated in Fig. 4c, which shows that the distance between 2-butene and Pd surface decreases from 3.162 Å to 3.142 Å, with an increase of $E_a$ to -2.08 eV. At distance below 3.142 Å, 2-butene was fully hydrogenated to butane. Comparing the $E_a$ of the semi-hydrogenation (-0.12 eV) and full hydrogenation (-2.08 eV) with the empirical value ($E_a$ -0.8 eV) for the spontaneous catalytic reaction running at room temperature (25-30 °C), it allows to conclude that the semi-hydrogenation of butadiene could run by

following the E-R model, while the full hydrogenation of 2-butene to butane does not follow the E-R model. Moreover, Supplementary Fig. 13 shows a supplementary catalytic reaction, using alkenes (i.e., 2-butenes and propene) as feedstocks, it was found that no full-hydrogenation of alkenes to alkanes (i.e., butane and propane) occurs in the developed electrochemistry-assisted process.

Figure 4b shows butadiene hydrogenation reaction with hydrogen atom settling at the interstitial site of Pd (111) facet. It is consistent with that in our electrochemistry-assisted hydrogenation process, as hydrogen atom prefers to stay at the interstitial site on Pd (111) surface after longitudinal diffusion in palladium membrane (e.g., XRD analysis in Supplementary Fig. 7). The simulation results show that the close-

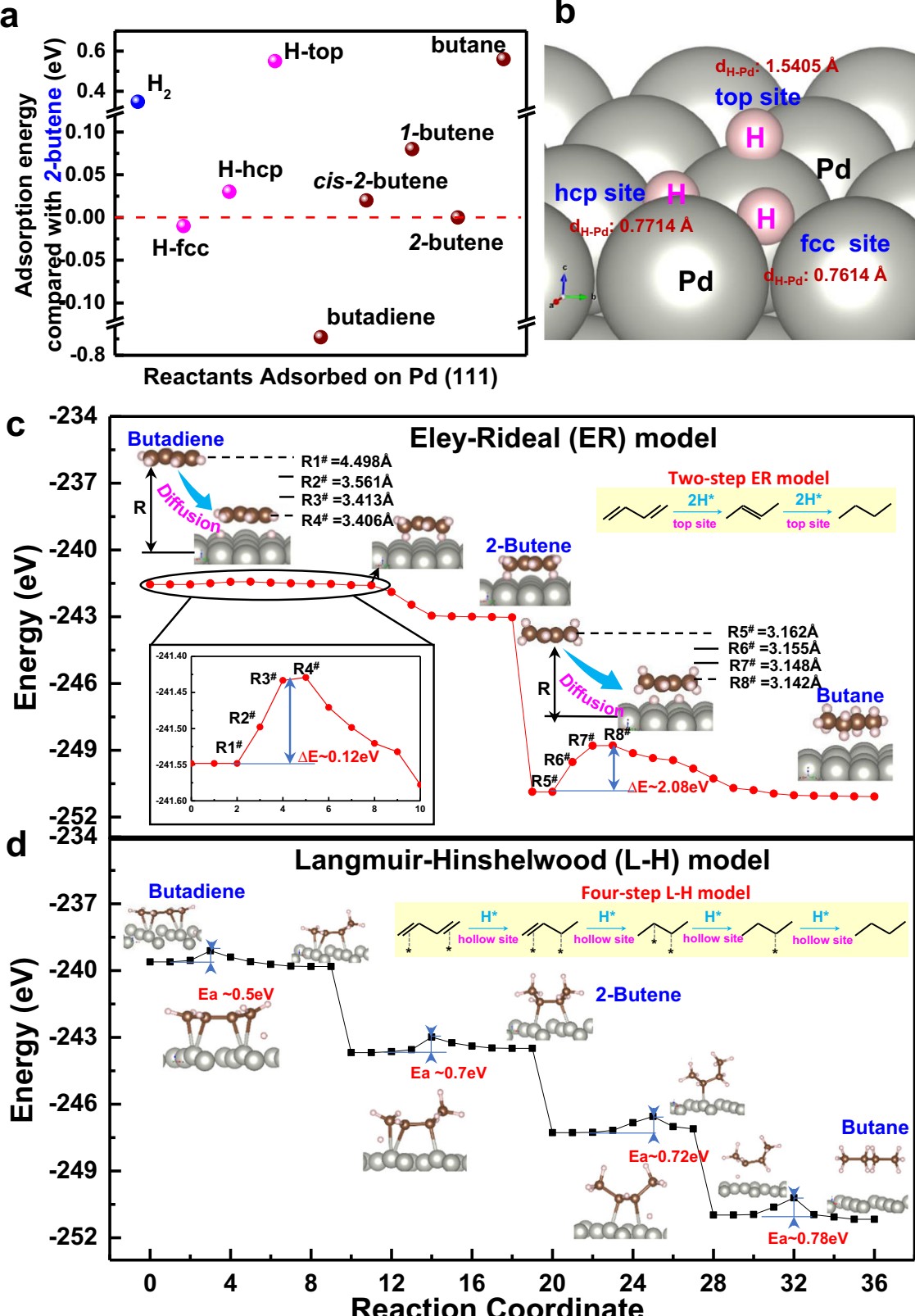

**Fig. 4 | DFT calculation on the energy evolution in butadiene hydrogenation. a** The adsorption energy of reaction species with 2-butene as the baseline on Pd (111) surface. **b** the state of H on different sites of Pd (111). **c** The Eley-Rideal model of reaction. **d** The Langmuir-Hinshelwood model of reaction.

packing of hydrogen atom with Pd atom generates a steric-hindrance from Pd atom to the catalytic reaction between adsorbed hydrogen atoms and gas-phase butadiene molecules. This leads to the conclusion that butadiene molecules must be adsorbed on Pd surface before

reacting with the adsorbed hydrogen atoms (i.e., the L-H model). As seen in Fig. 4c, the adsorbed butadiene reacts with hydrogen atoms ($H_a$), resulting in $CH_2 = CH\text{-}CH^{·}\text{-}CH_3$ as intermediate and then adsorbed 2-butene as semi-hydrogenation product, with Ea of ~0.5 eV and 0.7 eV,

respectively. Moreover, the adsorbed 2-butene was further hydrogenated to $CH_3$-$CH^{\cdot}$-$CH_2$-$CH_3$ and finally to butane, with Ea of -0.72 eV and 0.78 eV, respectively. It shows that the catalytic reaction could automatically run at room temperature (25-30 °C), following the L-H model.

Figure 4c, d reveals that the state of hydrogen atoms on Pd surface decides the catalytic reaction model of butadiene semi-hydrogenation, i.e., butadiene semi-hydrogenation follows the E-R model over H-rich Pd surface, and it runs following the L-H model with hydrogen atom staying in the interstitial site of Pd (111) facet. Full-hydrogenation of 2-butene to butane is independent from the state of hydrogen atom on Pd (111) facet, and it runs through the L-H model. Moreover, these calculations reveal that the conventional thermocatalytic route and the electrochemistry-assisted hydrogenation process follow different models. Specifically, the conventional thermocatalytic route with an excessively premixed $H_2$ offers palladium catalyst with a H-rich surface. Even though it promotes a rapid semi-hydrogenation of butadiene ($E_a$-0.12 eV), the rapid formation of butenes causes an excess of butenes accumulated on Pd surface, especially at high butadiene conversion, as the adsorption of butanes is stronger than $H_2$ and H-top. The adsorbed butenes either form oligomers by polymerization reaction, leading to the deactivation of Pd catalysts[29], or further hydrogenate to butane, offering the catalyst a low alkenes selectivity at high butadiene conversion. However, in our developed electrochemistry-assisted process, hydrogen atoms obtained from the longitudinal diffusion in Pd membrane for hydrogenation reaction mainly stay at the fcc site (i.e., H-fcc) of Pd surface (Fig. 4b). Their relatively low reactivity in butadiene hydrogenation reaction (Ea -0.7 eV) offers a slow formation of 2-butene products. Moreover, 2-butene easily desorb away from the Pd surface owing to its higher adsorption energy than that of hydrogen atom at the interstitial hole. Therefore, our electrochemistry-assisted hydrogenation process with water offers high catalytic selectivity to butenes, as well as high catalytic stability at room temperature (25-30 °C).

## Discussion

We developed an innovative room temperature (25-30 °C) electrochemistry-assisted selective hydrogenation process for the industrial elimination of alkynes and alkadienes gas impurities in an excess of alkene, using active hydrogen atom intermediates obtained from water electrolysis. For the proof of concept, the as-received commercial bulk Pd membrane was employed as the catalyst. The developed process offers above 97% of butadiene conversion and over 92% of alkenes selectivity in an excess of propene for over 360 h of reaction. Moreover, the developed process has a global energy consumption of 0.003 Wh/$mL_{butadiene}$, which is thousands of times lower than that of the current thermocatalytic route, and avoids the utilization of hydrogen gas. This process could be very attractive to bring about an evolution in industrial hydrogenation process owing to its highly-efficient catalytic performance, low energy consumption and high safety. To make the developed hydrogenation process more acceptable for industry, future research could be performed on developing low-cost Pd-based membrane (e.g., ultrathin self-supporting Pd membrane, Pd-based bimetallic membrane and etc), with a high hydrogen atom penetration and an enhanced catalytic hydrogenation performance.

## Methods
### Materials
Palladium membrane (purity: 99.99%, surface area: $100 \times 100$ mm, thickness: 25 µm) and Pt wire (diameter: 2 mm) were purchased from Alfa Aesar. The reference electrodes (i.e., Ag/AgCl, R0303) and the counter electrodes (i.e., Graphite rod) were purchased from the Tianjin Aida Hengsheng Technology Development Co., Ltd. Anhydrous sulfuric acid ($H_2SO_4$, purity: 98%) and hydrogen peroxide ($H_2O_2$, purity: 30 wt. % in $H_2O$) were obtained from the Sinopharm Group Chemical Reagent Co., Ltd. Potassium hydroxide (KOH, purity: 85%) and nitric acid ($HNO_3$, purity: 68% in $H_2O$) were obtained from the Aladdin Reagent Shanghai Co., Ltd. Commercial Pd/$Al_2O_3$ catalyst with 10 wt% of Pd loading was purchased from Alfa Aesar.

Palladium membrane was cut to the size of $20 \times 20$ $mm^2$. It was treated by immersing in an aqueous solution containing $HNO_3$, $H_2O$, and $H_2O_2$ with volume ratio 0.5: 0.5: 1. After 45 min, the palladium membrane was washed with deionized water three times, and then dried in vacuum at room temperature (25-30 °C).

### Characterization of the Pd membrane
XRD analysis was carried out on a D8 Bruker X-ray diffractometer, using the Cu Kα radiation (1.5418 Å; 40 kV and 30 mA) with a Ni filter and under air. Data were recorded at a scanning step of 0.05° with scanning range from 5°–90°. Scanning electron microscopy (SEM) images were acquired on Hitachi S-4800 scanning electron microscope at an accelerating voltage of 5.0 kV and an operating current of 10 µA, using a through-lens detector in secondary electron mode. A Bruker atomic force microscope (AFM) was used in air and contact modes to measure the roughness and to image surface morphology of the membrane. Transmission electron microscopy (TEM) images were acquired on a JEOL 2100 microscope (Japan) operating at an acceleration voltage of 200 kV.

### The electrochemical property of the Pd membrane
Electrochemical measurements were carried out in a typical three-electrode electrolytic cell using an Autolab (CS350H) purchased from Wuhan CORRTEST Instruments Company, with the basic electrochemical property of Pd membrane shown in Supplementary Fig. 5. The cleaned palladium membrane with an area of 4 $cm^2$ was used as the working electrode (W.E), Ag/AgCl electrode as the reference electrode (R.E) and a graphite rod as the counter electrode (C.E) in either 0.2 M KOH or 0.1 M $H_2SO_4$. Cyclic voltammetry scan was performed on the palladium membrane, with a scan rate of 10 mA $s^{-1}$. The voltage range (vs. Ag/AgCl electrode) was from −0.32 V to 1.75 V for 0.1 M $H_2SO_4$ and from −1.08 V to 0.5 V for 0.2 M KOH, respectively. The electrocatalytic properties of the palladium membrane in hydrogen evolution reaction (HER) in acidic electrolyte (0.1 M $H_2SO_4$) and in oxygen evolution reaction (OER) in basic electrolyte (0.2 M KOH) were evaluated by linear sweep voltammetry test, with a rate of 5 mV $s^{-1}$. The voltage ranges (vs. Ag/AgCl electrode) selected for linear sweep voltammetry test were from −0.9 V to 0.15 V for HER in 0.1 M $H_2SO_4$ and from 0 V to 1.0 V for OER in 0.2 M KOH, respectively.

### Study of the hydrogen atom transportation in the Pd membrane
The transportation behavior of the hydrogen atoms in the membrane was measured by hydrogen atom electrochemical titration (HAET) method, using a five-electrode electrochemical system in a H-cell[30] (Supplementary Fig. 14a). The principle of the HAET method is that hydrogen atoms are formed from water electrolysis in one cell under constant current density ($D_{i1}$), then they crossover the palladium membrane and finally are electro-oxidized in another cell, with the generation of an oxidation current ($D_{i2}$). $D_{i2}$ is directly related to the number of hydrogen atoms that cross over the palladium membrane. Specifically, the H-cell contains two single cells, which have a window (area of 2.0 $cm^2$) for assembling. Each single cell was equipped with one Ag/AgCl reference electrode (R.E) and one graphite rod counter electrodes (C.E). A compact Pd membrane was sealed between the two single cells, and it acts as a working electrode for both cells, i.e., as cathode in the hydrogen generation cell and as anode in the hydrogen oxidation cell. The cell for hydrogen generation was filled with 40 mL of 0.1 M $H_2SO_4$, while the hydrogen oxidation cell was filled with 40 mL of 0.2 M KOH. In order to avoid the side reaction of hydrogen evolution at the surface of the Pd membrane in the side of the hydrogen generation cell, a very low $D_{i1}$ range (−0.05 mA $cm^{-2}$ to 1.0 mA $cm^{-2}$)

was set for hydrogen atom generation by water electrolysis[31]. Moreover, an overpotential of 0.15 V was applied in the hydrogen oxidation cell. All the parameters related to the current density and the overpotential were chosen on the basis of the electrochemical property data (Supplementary Fig. 5). They had to be close to the double electrode layer region, in which no Faradaic reaction occurs. Before the HAET test, the residual hydrogen atoms in the Pd membrane were firstly eliminated by electro-oxidation, i.e., $D_{i2}$ reached below 2 µA cm$^{-2}$ with keeping $D_{i1}$ at 0 mA cm$^{-2}$. Then, different $D_{i1}$ between −0.05 and −1 mA cm$^{-2}$ were applied for recording the evolution of $D_{i2}$ as a function of the diffusion time. After 400 s, $D_{i2}$ reached a constant irrespective of the $D_{i1}$ value. Then $D_{i1}$ was set back to 0 mA cm$^{-2}$. The HAET test ended when $D_{i2}$ decreased below 2 µA cm$^{-2}$.

## Catalytic evaluation

The scheme of the electrochemistry-assisted selective hydrogenation system is depicted in Fig. 1 and Supplementary Fig. 1a, b. As shown in Fig. 1a, the setup consists of a gas reactants cylinder, a compartment containing the liquid electrolyte with a circulating loop and the electrochemical-assisted reactor (Fig. 1b). The reactor has a sandwich structure with a gas feed cell and a liquid cell separated by a dense palladium membrane of 4 cm$^2$. In the liquid cell, an Ag/AgCl reference electrode (R.E.) and a graphite rod counter electrode (C.E.) were sealed. The 0.1 M H$_2$SO$_4$ electrolyte was circulated in the cell through a pump. The palladium membrane acted as the working electrode (W.E.). The surface of the palladium membrane available for water electrolysis was 0.785 cm$^2$, as a result of the use of the seal ring (inside diameter 1 cm) on the device. The gas-phase cell has the structure of a fixed bed reactor with the palladium membrane as catalyst. In this side, there was no seal ring, so the available surface of the membrane was 1.131 cm$^2$. The flow rate of gas reactant (e.g., 0.6% of butadiene in helium, 0.3% of acetylene in 3% propene, 0.6% of 2-butene in helium, and 3% of propene in helium) was controlled by a flowmeter. All the catalytic tests were performed at ambient temperature (25-30 °C). The composition of the catalytic products was in-situ analyzed every 15 min by on line gas chromatography (GC2030Smart equipped with a flame ionization detector (FID) containing a 7.5 m column (1/8 in.) filled with sebaconitrile 25% Chromosorb PAW 80/100 mesh and a thermal conductivity detector (TCD) with bridge current of 70 mA). The temperatures of the column and FID were set at 75 and 220 °C, respectively. Furthermore, the outlet gas products were also analyzed by an online mass-spectroscopy (MS, Hiden HPR-40 DEMS from Hiden Analytical Co., Ltd.). Three parameters, namely the current density for hydrogen generation, the electrolyte flow rate, and the gas hourly space velocity (GHSV), were chosen to study their effect on the catalytic performances/properties. For the study of the influence of the current density, the latter was varied from −0.05 to 20 mA cm$^{-2}$ under electrolyte flow rate of 3 mL min$^{-1}$ and GHSV of 5308 h$^{-1}$. For the study of the influence of the electrolyte flow rate, the flow rate of 0.1 M of H$_2$SO$_4$ was varied from 0 to 5 mL min$^{-1}$ under different current densities, with butadiene GHSV of 5308 h$^{-1}$. The study on the influence of the butadiene flow rate was performed under different current densities and with electrolyte flow rate of 5 mL min$^{-1}$.

As for the catalytic stability, a gas mixture of butadiene (0.6%) in an excess of propene (6%), with helium as balance gas, was used as the reactant for the electrochemistry-assisted hydrogenation reaction. The reaction was performed under the current density of −15 mA cm$^{-2}$, the electrolyte flow rate of 5 mL min$^{-1}$ and the GHSV of 5308 h$^{-1}$. The electrolyte (0.1 M H$_2$SO$_4$ aqueous) was changed after every 24 h of reaction for over 360 h of reaction. For comparison, 1 mg of 10 wt% Pd/Al$_2$O$_3$ diluted by 95 mg of Al$_2$O$_3$ was filled in a U-shape tube reactor (volume at 0.17 mL), and reduced in-situ in hydrogen at 200 °C for 2 h, with a temperature rate of 1 °C min$^{-1}$. Then the catalytic stability test of this commercial catalyst was performed at room temperature (25-30 °C), with a gas mixture of 0.6% butadiene, 6% propene, and 20% hydrogen gas in helium with GHSV of 5308 h$^{-1}$.

**Definition of different terms**:

1. Gas hourly space velocity (GHSV) is the ratio between the volume of gas reactants and the volume of catalyst per hour (h$^{-1}$):

$$GHSV = \frac{v_{gas} \times time}{v_{catalyst}} \qquad (1)$$

Here, $v_{gas}$ is the flow rate of reactants (mL min$^{-1}$); time is 1 h (60 min); $v_{catalyst}$ is the volume of catalyst. Here, the volume of gas-phase reactor of 0.1131 mL was used as $v_{catalyst}$.

2. The butadiene conversion was calculated as follows:

$$Conversion_{butadiene}(\%) = 100 \times \left(1 - \frac{Butadiene_{outlet}}{Butadiene_{inlet}}\right) \qquad (2)$$

3. The catalytic activity was expressed as turnover frequency (TOF) based on the following formula:

$$TOF(s^{-1}) = \frac{n_{converted\ butadiene}}{\rho \times S} \qquad (3)$$

$$PV = n\,RT \qquad (4)$$

Here, $n_{converted\ butadiene}$ is the number of butadiene molecular converted on Pd surface per second, and it is calculated from formula (1), (2) and (4); $\rho$ is the metal surface site density (i.e., $1.27 \times 10^{15}$ per cm$^2$ for palladium); $S$ is the valid surface area of Pd membrane for catalytic reaction (i.e., 1.131 cm$^2$).

4. The catalytic selectivity to butenes was expressed as follows:

$$Selectivity\ to\ butenes(\%) = \frac{Butenes_{outlet}}{Butenes_{outlet} + Butane_{outlet}} \times 100\% \qquad (5)$$

5. The Faradaic efficiency for catalytic hydrogenation reaction was calculated as follows:

$$FE = \frac{Q_{butadiene\ hydrogenation}}{Q_{total}} \times 100\% \qquad (6)$$

$$Q_{total} = i \times t \qquad (7)$$

$$Q_{butadiene\,hydrogenation} = n_{butenes}*2 + n_{butane}*4 \qquad (8)$$

It is mainly based on the following reactions:

$$H^+ + e^- = H \qquad (9)$$

$$Butadiene + 2H \rightarrow Butenes \qquad (10)$$

$$Butadiene + 4H \rightarrow Butane \qquad (11)$$

where $Q_{butadiene\ hydrogenation}$ is the total electric consumption for butadiene hydrogenation reaction; It was calculated by the formula (8), and the relationship between electrons, hydrogen atom and butenes/butane was presented in the formula (9-10). $Q_{total}$ was calculated from the current ($i$) applied for water electrolysis, with the reaction time ($t$).

6. The energy consumption expressed in kW·h was calculated as follows:

$$W = Pt \qquad (12)$$

$$P = Ui \qquad (13)$$

where W is electric work; P is electric power; $t$ is time; $U$ is voltage and $i$ is current. In this study, the stability test was conducted under a current density of $-15\,mA\,cm^{-2}$ (i.e., with $i = 11.781\,mA$). The average voltage is 1.03 V in Supplementary Fig. 11 and the average value was used for electric power calculation.

## Density functional theoretical calculation

Energy calculations in this work were performed using the Vienna ab initio simulation package (VASP)[32,33] based on the density functional theory (DFT). The generalized gradient approximation (GGA-PW91) was chosen as the exchange-correlation functional[34]. The kinetic energy cutoff was set at 450 eV. The Brillouin zone sampling was treated using the Monkhorst-pack grid[35], and a $4 \times 4 \times 1$ Monkhorst-pack K-point mesh was used during the whole calculation. The optimization thresholds were $10^{-5}$ eV and 0.015 eV/Å for electronic and ionic relaxations, respectively. In all cases, the saddle point nature of the transition states was assessed by the calculation of the numeric Hessian with a step of 0.02 Å and its diagonalization that rendered a unique imaginary frequency. In addition, the Climbing Image-Nudged Elastic Band (CI-NEB) method was used for the determination of transition states[36] in butadiene hydrogenation reaction. Six intermediate images were constructed between the initial and final states. Transition states were checked by ensuring that they have a force tangent to the reaction coordinate of less than 0.05 eV/Å. The frequency analysis was further performed to confirm the located TSs[37,38].

## Models

The Pd (111) surface was constructed from the respective crystal with the DFT-optimized lattice constant of 3.939 Å. Pd (111) slab consists of four atomic layers with a $(3 \times 3)$ unit cell and separated by 15 Å vacuum space. For the adsorption of hydrogen atom, hydrogen atom was set on different sites, atop site (top), hollow site of face centered cubic-packed (fcc), hollow site of hexagonal closest-packed (hcp) and bridge site (bri) on the surface of Pd (111). The bridge (bri) site was identified as the transition state (TS) due to its unstable structure (Supplementary Fig. 6). The total number of hydrogen atoms for monolayer (ML) adsorption was 27, a summation of all the number of fcc, hcp and top sites. Models with different hydrogen coverages on Pd (111) (from 0.037 to 0.778 ML) were built to evaluate the adsorption energy evolution as a function of the hydrogen coverage. To simulate the adsorption energy on surface defect of Pd (111), hydrogen atoms with a fixed $x$ and $y$ axis were introduced to the site near single Pd site. Moreover, considering the fact that Pd is not single crystal, here, the slab of Pd(335), containing six atomic layers with a $(3 \times 1)$ unit cell and separated by 15 Å vacuum space, was built for further calculation on the energy evolution of butadiene hydrogenation over Pd(335) facet.

The surface energy ($\sigma$), adsorption energy ($E_{ads}$), activation energy ($E_a$), and the total energy change ($\triangle E$) were calculated as follows:

$$\sigma = \sigma^{unrel} + E^{rel} \tag{14}$$

$$\sigma^{unrel} = \frac{1}{2}(E_{slab}^{unrel} - N_{atoms} \times E_{bulk}) \tag{15}$$

$$E^{rel} = E_{slab}^{unrel} - E_{slab}^{rel} \tag{16}$$

in which, $E_{slab}^{unrel}$ and $E_{slab}^{rel}$ are the energies of slab before and after structure relaxation; $N_{atoms}$ is the number of palladium atom in the slab; $E_{bulk}$ is the energy of single palladium atom in bulk state.

$$E_{ads} = E_{slab}^{ads} - E^{ads} - E_{slab} \tag{17}$$

$$E_a = E_{TS} - E_{IS} \tag{18}$$

$$\triangle E = E_{FS} - E_{IS} \tag{19}$$

where, $E_{slab}^{ads}$, $E^{ads}$ and $E_{slab}$ are the energies of calculated adsorbates-slab, adsorbates in gas-phase and pure slab, respectively; $E_{FS}$, $E_{TS}$ and $E_{IS}$ are the energies of final states, transition states, and initial states, respectively.

## Data availability

All data are available within the Article and Supplementary Files, or available from the corresponding authors on reasonable request. Source data are provided with this paper.

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

## Acknowledgements

This work was supported by the Program of Introducing Talents of Discipline to Universities-Plan 111 (B0002) from the Ministry of Science and Technology and the Ministry of Education of China. We thank the National Natural Science Foundation of China (grant 21902122, 22293020, and 22293022), the National Key R&D Program of China (grant 2021YFE0115800), the Postdoctoral Science Foundation of China (grant 2019M652723) and Foshan Xianhu Laboratory of the Advanced Energy Science and Technology Guangdong Laboratory (XHD2020-002). We thank Professor Jun-Sheng Li for providing the utilization permission of VASP resource. The work was carried out at LvLiang Cloud Computing Center of China, and the calculations were performed on TianHe-2. We thank Professor Hazar Guesmi for her advice on DFT calculation and analyzation and Professor Catherine Louis for her advice on catalytic results analyzation.

## Author contributions

B.L.S. and Z.W. supervised the work. Z.W., Y.Q.Y., and Y.C. performed the experiments and analyzed the data. Z.W. and H.L.T. worked on all computational investigations. Z.W., L.H.C., and B.L.S. co-wrote the paper. All authors discussed the results and commented on the manuscript. Z.W. and B.L.S. finalized the manuscript.

## Competing interests

The authors declare no competing interests.
