## [Peer Review File · Nature Communications]

REVIEWER COMMENTS

Reviewer #1 (Remarks to the Author):

This is a useful and well written report describing an innovative room temperature electrochemistry-assisted selective hydrogenation process for the elimination of butadiene gas impurity in an excess of propene, using hydrogen atoms in-situ generated by water electrolysis. The present process offers excellent butadiene conversion and selectivity of semi-hydrogenated products even in an excess of propene after a long reaction time (360 h). In addition, the global energy consumption evaluated for this process is several thousand times lower than that of the conventional thermocatalytic route.

The work is novel, timely, and original. The manuscript is well written with good quality figures and table. The SI section is also useful. Experimental methods are explained. The literature overview is appropriate. The title and abstract are consistent with the main work.

Overall, the report is very strong and would make a nice contribution to Nature Communications. I recommend that it be published after the authors consider the following comments:

- 1) The thickness of the Pd membrane is expected to have a significant effect on the efficiency of the hydrogenation reaction. Is the 25 μm film thickness used in this work the optimal thickness? A discussion of the effect of the thickness of the Pd membrane is also needed.
- 2) In Fig. 2a, the FE decreases with increasing current density, reaches a minimum at 5 mA cm^{-2} , and then increases again with increasing current density. A more detailed interpretation of this behavior should be given in the main text.
- 3) In page 12, line 216: The authors claim that the present process can keep I-1_2 away from butadiene reactants. However, there is concern that H_a permeating through the Pd membrane may also couple at the membrane interface on the gas phase side to form molecular hydrogen. Therefore, it is necessary to investigate whether E^+ contamination is really impossible in the gas phase.
- 4) The authors explain the factors responsible for the selective formation of 2-butene under high butadiene conversion with the LI-I model based on DFT calculations. Certainly, this explanation is reasonable, but to support it, it should also be shown that no hydrogenation of 2-butene occurs even when 2-butene is used as a starting material.
- 5) If possible, the process should also be experimentally verified with alkynes as impurities.

Reviewer #2 (Remarks to the Author):

This manuscript describes the room-temperature electrochemical-assisted semi-hydrogenation of butadiene to butane using H_2O as the hydrogen source. An innovative gas-feed fixed bed reactor, comprising a commercial palladium membrane sandwiched between

a liquid-phase cell and a gas-phase cell, has been specifically designed to carry out the aforementioned chemical transformation in continuous flow conditions. The interest of the process aimed to reduce remaining alkynes/alkadienes impurities from alkenes feedstocks, which are widely used for polymerization reactions, is well explained in the introduction. The performance of the designed catalytic system in terms of activity, stability, and alkene selectivity for the conversion of butadiene, even in the presence of a large excess of propene, is impressive suggesting that the reported procedure will surely find applicability. Moreover, it has been demonstrated that, compared to the most common Pd-catalyzed thermocatalytic process, the herein reported electrochemical-assisted one is more selective and energy-saving. In addition, the manuscript also present complementary DFT calculations, which have been used to explain the high Faradaic efficiency towards the desired electrochemical-assisted hydrogenation reaction, as well as the origin of the selectivity and catalyst stability. As a summary, I think this is an excellent piece of work and it is remarkably well written, so I recommend publication in Nature Communications.

Reviewer #3 (Remarks to the Author):

In the present work, the authors report a room temperature electrochemistry-assisted selective hydrogenation process in gas-feed fixed bed reactor with water. The performance of Pd membrane catalyst is significant high for selective butadiene hydrogenation with alkenes selectivity staying around 92% at butadiene conversion above 97%. There are still some serious problems of the paper for the authors in the consideration.

1. The present work lack of characterization of the catalyst for the further reaction mechanism investigation.
2. What is the structures of Pd catalyst under the reaction conditions?
3. Without solvent effect, it is hardly for the authors to derive the reaction mechanism of butadiene hydrogenation correctly.
4. The authors have revealed the reaction mechanisms of butadiene hydrogenation by DFT calculations. Can the DFT calculations results be comparable with the experimental measurements?

Responses to Reviewer's report

To Reviewer # 1

Reviewer #1 (Remarks to the Author):

This is a useful and well written report describing an innovative room temperature electrochemistry-assisted selective hydrogenation process for the elimination of butadiene gas impurity in an excess of propene, using hydrogen atoms in-situ generated by water electrolysis. The present process offers excellent butadiene conversion and selectivity of semi-hydrogenated products even in an excess of propene after a long reaction time (360 h). In addition, the global energy consumption evaluated for this process is several thousand times lower than that of the conventional thermocatalytic route.

The work is novel, timely, and original. The manuscript is well written with good quality figures and table. The SI section is also useful. Experimental methods are explained. The literature overview is appropriate. The title and abstract are consistent with the main work. Overall, the report is very strong and would make a nice contribution to Nature Communications. I recommend that it be published after the authors consider the following comments:

- 1) The thickness of the Pd membrane is expected to have a significant effect on the efficiency of the hydrogenation reaction. Is the 25 μm film thickness used in this work the optimal thickness? A discussion of the effect of the thickness of the Pd membrane is also needed.
- 2) In Fig.2a, the FE decreases with increasing current density, reaches a minimum at 5 mA cm^{-2} , and then increases again with increasing current density. A more detailed interpretation of this behavior should be given in the main text.
- 3) In page 12, line 216: the authors claim that the present process can keep H_2 away from butadiene reactants. However, there is concern that H_a permeating through the Pd membrane may also couple at the membrane interface on the gas phase side to form molecular hydrogen. Therefore, is it necessary to investigate whether H_2 contamination is really impossible in the gas phase.
- 4) The authors explain the factors responsible for the selective formation of 2-butene under high butadiene conversion with the LH model based on DFT calculations. Certainly, this explanation is reasonable, but to support it, it should also be shown that no hydrogenation of 2-butene occurs even when 2-butene is used as starting material.
- 5) If possible, the process should also be experimentally verified with alkynes as impurities

Answer: We thank you for your positive comments and your constructive inputs and suggestions, which are very helpful in improving the quality of our manuscript. We have carefully and thoroughly revised the manuscript by taking each of your suggestions and comments into account by carrying out a series of supplementary experiments to improve the quality of our manuscript. We do hope that our supplementary experiments, explanations and thorough revisions based on your questions and your professional advice will satisfy you.

Please find below our point-by-point answers to your comments and suggestions. *Your comments and questions are written in blue italics* and our responses in “black”. For your facility, we have 1) shown at the end of our answer to each of your questions the changes we have made in the revised manuscript and supplementary information with page number and the numbers of lines and 2) also highlighted **in red colour** all the changes we have made in the revised manuscript and supplementary information.

General comment:

This is a useful and well written report describing an innovative room temperature electrochemistry-assisted selective hydrogenation process for the elimination of butadiene gas impurity in an excess of propene, using hydrogen atoms in-situ generated by water electrolysis. The present process offers excellent butadiene conversion and selectivity of semi-hydrogenated products even in an excess of propene after a long reaction time (360 h). In addition, the global energy consumption evaluated for this process is several thousand times lower than that of the conventional thermocatalytic route.

The work is novel, timely, and original. The manuscript is well written with good quality figures and table. The SI section is also useful. Experimental methods are explained. The literature overview is appropriate. The title and abstract are consistent with the main work. Overall, the report is very strong and would make a nice contribution to Nature Communications. I recommend that it be published after the authors consider the following comments:

Answer: Thank you very much for your positive comment. We have modified the manuscript to take very carefully your specific comments into account.

Specific comments:

- 1. The thickness of the Pd membrane is expected to have a significant effect on the efficiency of the hydrogenation reaction. Is the 25 μm film thickness used in this work the optimal thickness? A discussion of the effect of the thickness of the Pd membrane is also needed.*

Answer: Thank you for your good question and very helpful suggestion.

We highly agree with your opinion that the thickness of Pd membrane could have a significant effect on the efficiency of the hydrogenation reaction. The process of electrochemistry-assisted

hydrogenation reaction (**Figure R1a**) shows that the penetration of hydrogen atoms (H_a) in Pd membrane is one of the key factors for deciding the catalytic hydrogenation reaction. The utilization of 25 μm thickness of Pd membrane for the proof of concept is mainly due to its easily accessible in market.

In fact, the penetration of H_a in Pd membrane obeys the Fick's first law (**Figure R1b**).

Figure R1. (a) The process of electrochemistry-assisted hydrogenation reaction presented in Figure 2d; (b) The formula of the Fick's first law;

In **Figure R1b**, J_H is the penetration flux of H_a ; D is penetration coefficient of H_a in palladium; ∂C_H is the H_a concentration gap along the thickness direction of Pd membrane; x is the thickness of Pd membrane. Therefore, the thinner thickness (i.e., smaller x) of Pd membrane, the larger J_H under a constant current density for water electrolysis is, resulting in an enhanced catalytic performance.

On the other side, the mechanical strength and compactness of self-supported Pd membrane are also very important as the self-supported Pd membrane has to be assembled into the reactor for separating reactants gas and electrolyte liquid phase. Therefore, the optimal thickness of self-supported Pd membrane has to take both the H_a diffusion flux and the mechanical property into consideration.

It is worth to say that the immature technology for producing self-supported Pd membrane prevents us to obtain Pd membrane with different thickness. The thinner thickness of self-supporting Pd membrane, the more serious problem of mechanical strength and thickness uniformity is. We obtained a Pd membrane with 8 mm thickness from commercial supplier. Unfortunately, the quality of this membrane is bad with many surface wrinklies and some holes and the mechanical strength is not good enough to be assembled into the reactor. Pd membrane 25 μm thickness as the common benchmark product has been used in our study.

Following your valuable suggestion, the effect of the thickness of Pd membrane on the efficiency of hydrogenation reaction has been discussed in the revised manuscript **at line 147 - 152 in page. 8**, as follows:

“The thickness of Pd membrane has a significant effect on the efficiency of the hydrogenation reaction. Thinner Pd membrane can decrease the longitudinal diffusion of distance and accelerate H_a penetration through Pd membrane, thus enhance the efficiency of hydrogenation reaction. However, the mechanical strength and compactness of self-supported Pd membrane should also be taken into account.”

We hope this detailed discussion can satisfy you.

2. *In Fig.2a, the FE decreases with increasing current density, reaches a minimum at 5 mA cm⁻², and then increases again with increasing current density. A more detailed interpretation of this behavior should be given in the main text.*

Answer: Thank you for your very valuable suggestion.

Following your suggestion, a more detailed interpretation of the evolution of FE as a function of current density, especially at around 5mA cm⁻², has been added in the revised manuscript, **page 7-8, lines 119~141.**

“In the reaction process, D_i solely decides the formation of H_a at the surface of Pd membrane in liquid cell, and the formed H_a could either penetrate through Pd membrane obeying the Fick’s first law for butadiene hydrogenation (i.e., contributing to the FE), or combine to form H₂ as the self-supported Pd membrane has to be assembled into the reactor for separating reactants gas and electrolyte liquid phase. Detailed discussion on the H_a penetration in Pd membrane under various D_i was summarized in the **Supplementary Discussion Part 1**. It was found that, at low D_i (e.g., <0.15 mA cm⁻², without H₂ evolution from the CV analysis in **Supplementary Fig. 5**), hydrogen atoms first prefer to laterally diffuse within the atomic layer of palladium lattice and then longitudinally permeate through the palladium membrane owing to its low energy barrier of diffusion (i.e., 0.16 eV in **Supplementary Fig. 6**). It offers a high utilization of H_a (i.e., high FE at low D_i). With the increase of D_i offering a rapid H_a formation and its accumulation on Pd surface, hydrogen atoms tend to directly penetrate palladium lattice and vertically pass through Pd membrane for catalytic reaction (**Supplementary Discussion Part 1**). Simultaneously, the increased D_i promotes H₂ formation on the interface of Pd membrane to the electrolyte liquid phase, and the formed H₂ decrease the percentage of H_a for butadiene hydrogenation. Specially, a phase transformation from Pd to PdH_{0.75} was identified when D_i above -5mA cm⁻² (**Figure R2a**), which shows a swelled palladium lattice from 3.89Å of Pd to 4.02 Å of PdH_{0.75}. The interstitial hole in between octahedron void and tetrahedral void in Pd crystal increases from 0.217 Å to 0.315 Å (**Figure R2b**), which is larger than the diameter of hydrogen atom (0.25 Å), offering a continuous H_a penetration. Therefore, the FE rapidly increases from 11.5% to 52.3% with D_i rising from 5mA cm⁻² to 15mAcm⁻².”

Figure R2. (a) XRD of Pd membrane after catalytic reaction in **Supplementary Fig. 7**; (b) Crystal structure of PdH_{0.75}, with a calculation on the interstitial hole in PdH_{0.75} in **Supplementary Fig. 16c**

We hope this detailed discussion can satisfy you.

3. *In page 12, line 216: the authors claim that the present process can keep H₂ away from butadiene reactants. However, there is concern that H₂ permeating through the Pd membrane may also couple at the membrane interface on the gas phase side to form molecular hydrogen. Therefore, is it necessary to investigate whether H₂ contamination is really impossible in the gas phase.*

Answer: Thank you for your excellent suggestion.

We agree with your opinion that hydrogen atoms permeating through the Pd membrane have the possibility to couple at the membrane interface on the gas phase side to H₂ as the process is widely used for H₂ purification in industry. Following your suggestion, we have investigated this possibility by both in-situ gas chromatography (GC) coupled with in-situ mass-spectroscopy (MS) analysis in **Figure R3** and the DFT calculation (**Supplementary Fig. 9**).

As for the in-situ GC coupled with MS analysis on the outlet gas from reactor in **Figure R3**, the hydrocarbon products checked by the FID detector of GC (**Figure R3b**) shows above 90% of butadiene conversion under 15 mA cm⁻² of water electrolysis and the GHSV of butadiene ~5308h⁻¹. Meanwhile, the possible H₂ product in the outlet gas was evaluated by the TCD detector of GC in **Figure R3c**, which shows that no H₂ is formed. In addition, the online MS analysis was employed on the outlet gas in **Figure R3d**, and it verifies that the main products are hydrocarbons.

Figure R3. The digital curve of GC-MS data during catalytic reaction. (a) scheme of analysis; (b) FID analysis; (c) TCD analysis; (d) MS analysis;

Moreover, our DFT calculation in **Supplementary Fig. 9** also predicts that the adsorbed hydrogen atoms (H_a) on Pd membrane in the gas-phase cell prefer to react with butadiene and butenes for their hydrogenations, rather than combines to form H_2 , as the activation energy (E_a) for hydrogenation is ~ 0.8 eV, which is much lower than that of H_a combination ($E_a \sim 1.09$ eV). Therefore, both DFT calculation and in-situ mass-spectroscopy (MS) analysis clearly demonstrate that the H_a prefers to react with butadiene, rather than combines to form H_2 during the semi-hydrogenation reaction.

We have added the description in the main manuscript (**page 11, lines 221-222**).

“i.e., verification by the in-situ GC coupled with on-line mass-spectroscopy analysis on the outlet gas products (**Supplementary Fig. 12**)”

We hope this detailed discussion can satisfy you.

4. *The authors explain the factors responsible for the selective formation of 2-butene under high butadiene conversion with the LH model based on DFT calculations. Certainly, this explanation is reasonable, but to support it, it should also be shown that no hydrogenation of 2-butene occurs even when 2-butene is used as starting material.*

Answer: Thank you for your very helpful suggestion.

Following your valuable suggestion, we have performed supplementary hydrogenation experiments by using alkenes (e.g., 6000 ppm of 2-butene in He and 30000 ppm of propene in

He) as starting materials in **Figure R4**, which shows that no alkanes (e.g., butane and propane) formation from hydrogenation of alkenes.

Therefore, the supplementary experiment shows no full-hydrogenation of alkenes to alkanes during catalytic reaction.

We have added a detailed description in our revised manuscript (**page 13, line 266-268**).

“Moreover, supplementary catalytic reaction using alkenes (i.e., 2-butenes and propene) as feedstocks, no full-hydrogenation of alkenes to alkanes (i.e., butane and propane) was observed in **Supplementary Fig. 13**.”

Figure R4. The digital curve of GC data during catalytic reaction with various alkenes feedstocks under different current densities at the GHSV of $\sim 5308\text{h}^{-1}$. (a) 0.6% of 2-butene in helium; (b) 3% of propene in helium;

We hope this detailed discussion can satisfy you.

5. *If possible, the process should also be experimentally verified with alkynes as impurities*

Answer: Thank you for your valuable suggestion.

Following your precious suggestion, we have performed supplementary hydrogenation experiments by using 3000 ppm of acetylene as impurity in 3% of propene with helium as balance in **Figure R5**.

Figure R5 shows that the acetylene conversion gradually increases from 0% to above 70% in gas-phase cell with an increase of current density for water electrolysis from 0 to 15 mA cm^{-2} . It has a very similar catalytic behavior as that of butadiene. Therefore, the relatively high catalytic performance for acetylene semi-hydrogenation in an excess of propene indicates that the developed electrochemistry-assisted hydrogenation process has unique selectivity to semi-hydrogenation, high reliable and reproducibility.

Figure R5. The digital curve of GC data of acetylene (3000ppm) semi-hydrogenation in an excess of propene (3%) under different current densities at the GHSV of $\sim 5308 \text{ h}^{-1}$.

We have added a detailed description in our revised manuscript (**page 5, line 98-99**).

“Similar catalytic behavior was also obtained by using 0.3% of acetylene in an excess of 3% propene with helium as balance in **Supplementary Fig. 4**.”

We hope this detailed discussion can satisfy you.

Finally, we would like to thank this reviewer for his/her precious time and professionalism devoted to the evaluation of our manuscript. His/her valuable and constructive comments indeed improve the quality of our manuscript.

To Reviewer # 2

Reviewer #2 (Remarks to the Author):

This manuscript describes the room-temperature electrochemical-assisted semi-hydrogenation of butadiene to butane using H₂O as the hydrogen source. An innovative gas-feed fixed bed reactor, comprising a commercial palladium membrane sandwiched between a liquid-phase cell and a gas-phase cell, has been specifically designed to carry out the aforementioned chemical transformation in continuous flow conditions. The interest of the process aimed to reduce remaining alkynes/alkadienes impurities from alkenes feedstocks, which are widely used for polymerization reactions, is well explained in the introduction. The performance of the designed catalytic system in terms of activity, stability, and alkene selectivity for the conversion of butadiene, even in the presence of a large excess of propene, is impressive suggesting that the reported procedure will surely find applicability. Moreover, it has been demonstrated that, compared to the most common Pd-catalyzed thermocatalytic process, the herein reported electrochemical-assisted one is more selective and energy-saving. In addition, the manuscript also present complementary DFT calculations, which have been used to explain the high Faradaic efficiency towards the desired electrochemical-assisted hydrogenation reaction, as well as the origin of the selectivity and catalyst stability.

As a summary, I think this is an excellent piece of work and it is remarkably well written, so I recommend publication in Nature Communications.

Answer: Thank you very much for your very positive comment.

We are very grateful to you for your very positive and encouraging comment. With your encouragement, we will continue to work hard to provide more innovant research works.

We would like to thank this reviewer for his/her precious time and professionalism devoted to the evaluation of our manuscript to give very positive comment to our manuscript.

To Reviewer # 3

Reviewer #3 (Remarks to the Author):

In the present work, the authors report a room temperature electrochemistry-assisted selective hydrogenation process in gas-feed fixed bed reactor with water. The performance of Pd membrane catalyst is significant high for selective butadiene hydrogenation with alkenes selectivity staying around 92% at butadiene conversion above 97%. There are still some serious problems of the paper for the authors in the consideration.

1. The present work lack of characterization of the catalyst for the further reaction mechanism investigation.
2. What is the structures of Pd catalyst under the reaction conditions?
3. Without solvent effect, it is hardly for the authors to derive the reaction mechanism of butadiene hydrogenation correctly.
4. The authors have revealed the reaction mechanisms of butadiene hydrogenation by DFT calculations. Can the DFT calculations results be comparable with the experimental measurements?

Answer: We are very grateful to you for your input and your precious suggestions which are indeed helpful in improving the quality of our manuscript. Thanks for your professionalism. We have performed a series of supplementary experiments and modified the manuscript to take very carefully your professional comments into account. We wish that our responses and our modifications on the basis of your comments can satisfy you.

Please find below our point-by-point answers to your comments and suggestions. *Your comments and questions are written in blue italics* and our responses in “black”. For your facility, we have 1) shown at the end of our answer to each of your questions the changes we have made in the revised manuscript and supplementary information with page number and the numbers of lines and 2) also highlighted **in red colour** all the changes we have made in the revised manuscript and supplementary information.

General comment:

In the present work, the authors report a room temperature electrochemistry-assisted selective hydrogenation process in gas-feed fixed bed reactor with water. The performance of Pd membrane catalyst is significant high for selective butadiene hydrogenation with alkenes selectivity staying around 92% at butadiene conversion above 97%.

Answer: Thank you very much for your pertinent comments. Thank you also for your specific suggestions that indeed help us to improve the quality of our manuscript.

We have modified the manuscript to take very carefully your comments into account.

Specific comments:

- 1. The present work lack of characterization of the catalyst for the further reaction mechanism investigation.*

Answer: Thank you for your very helpful suggestion.

In this study, the as-received bulky Pd membrane (10 cm*10 cm, thickness of 25 um, purity: 99.99%, Shanghai Rush Metal Co., Ltd) was employed as the probe catalyst for proofing a new concept of electrochemistry-assisted selective hydrogenation with water. Various characterizations have been carried on the Pd membrane, including the surface topography before and after hydrogenation (e.g., SEM, AFM in **Supplementary Fig. 3**), the crystal structure of Pd-based membrane before and after hydrogenation (e.g., XRD in **Supplementary Fig. 2** and **Supplementary Fig. 7**), the basic electrochemical property of the as-received Pd membrane for water electrolysis (e.g., CV and LSV in **Supplementary Fig. 5**), the hydrogen atom penetration behavior as a function of current density for water electrolysis in the as-received Pd membrane (e.g., the HAET test in **Supplementary Fig. 14**), and so on.

Specially, comparison with the crystal structure of as-received Pd membrane (face-centered cubic (fcc) structure, cell parameter of $3.89\text{\AA}\times 3.89\text{\AA}\times 3.89\text{\AA}$ and $90.0^\circ\times 90.0^\circ\times 90.0^\circ$) in **Supplementary Fig. 2**, XRD in **Supplementary Fig. 7** shows an appearance of palladium hydride ($\text{PdH}_{0.75}$, fcc structure, with a cell parameter of $4.02\text{\AA}\times 4.02\text{\AA}\times 4.02\text{\AA}$ and $90.0^\circ\times 90.0^\circ\times 90.0^\circ$) after electrochemistry assisted hydrogenation reaction. The slightly swelled palladium lattice was mainly due to the dissolution of hydrogen atoms (H_a) into the interstitial void of Pd crystal (**Supplementary Fig. 16**). This characterization shows that H_a in Pd lattice mainly stay in the interstitial void (e.g., the interstitial sites of face center cubic and/or the interstitial sites of hexagonal closest-packed). It offers a basic structure parameter of palladium hydride for further exploration on the reaction mechanism of electrochemistry-assisted hydrogenation reaction.

Moreover, the basic electrochemical property of the as-received Pd membrane was evaluated by the cyclic voltammetry scan (CV) and the linear sweep voltammetry scan (LSV) in **Supplementary Fig. 5**. They offer the key parameters (e.g., the potential vs. Ag/AgCl for proton electro-reduction to adsorbed H_a and the potential vs. Ag/AgCl for electro-oxidation of H_a at the interface of Pd membrane) for the exploration of H_a penetration in Pd membrane by the hydrogen atom electrochemical titration (HAET) method in **Supplementary Fig. 14**. The identified two different H_a penetrations highly agree with the DFT prediction in

Supplementary Fig. 6 and **Supplementary Fig. 16**, and support the discussion on the catalytic behavior as a function of the current density for water electrolysis in the liquid-phase cell.

Furthermore, in addition to the above various characterizations on Pd membrane, catalytic reactions with different types of H_a intermediates supplying (e.g., premixed butadiene with H_2 gas, H_2 promoted H_a penetration, and water electrolysis promoted H_a penetration) were also explored to deep analyze the proposed reaction mechanism of electrochemistry-assisted hydrogenation reaction in **Supplementary Fig. 8**. It further verifies the advantage of the innovative reaction process in the submitted work.

Therefore, we hope you can kindly agree that the reaction mechanism has been properly discussed based on these basic characterizations of Pd membrane.

2. What is the structures of Pd catalyst under the reaction conditions?

Answer: Thank you for your very helpful suggestion.

Following your helpful suggestion, we have checked the structure of Pd membrane after catalytic reaction by X-ray diffraction in **Figure R6a**, which shows a type of palladium hydride ($PdH_{0.75}$, face-centered cubic (fcc) structure, with a cell parameter of $4.02 \text{ \AA} \times 4.02 \text{ \AA} \times 4.02 \text{ \AA}$ and $90.0^\circ \times 90.0^\circ \times 90.0^\circ$) formation.

In our developed reaction process, hydrogen atoms are formed from water electrolysis in the liquid phase cell, then they penetrate in Pd membrane and finally react with butadiene on the contrary side of Pd membrane in the gas-phase cell. The XRD result in **Figure R6** shows that, comparison with the fresh Pd membrane (fcc structure, cell parameter of $3.89 \text{ \AA} \times 3.89 \text{ \AA} \times 3.89 \text{ \AA}$ and $90.0^\circ \times 90.0^\circ \times 90.0^\circ$), the crystal lattice of Pd was expanded to 4.02 \AA by dissolving hydrogen atoms into the interstitial void of Pd crystal, resulting in palladium hydride formation.

Figure R6. (a) XRD patterns of palladium membrane before and after electrochemistry assisted selective butadiene hydrogenation reaction with **(b)** corresponded palladium-based crystal structures in **Supplementary Fig. 7**.

We hope that you are satisfied with our answer to your questions on the structure of Pd catalyst under the reaction conditions.

3. *Without solvent effect, it is hardly for the authors to derive the reaction mechanism of butadiene hydrogenation correctly.*

Answer: Thank you for your very helpful suggestion.

After very careful analysis of your precious comment, in-depth discussion the electrochemistry-assisted selective hydrogenation process with several leading and internationally recognized colleagues in the field (they from France, USA, Canada, Japan, Belgium and China), all the authors and the above-mentioned colleagues did not understand what this reviewer would like to say about the “solvent effect” as the submitted work presents a gas/solid interface heterogeneous catalytic reaction in a gas-feed fixed-bed reactor. In detail, instead of using H₂ as hydrogen source, the present work obtains hydrogen atom intermediates from water electrolysis in a liquid phase cell on Pd membrane. The generated H_a intermediates on Pd membrane in the liquid-phase cell penetrate Pd membrane and arrive to the interface of Pd membrane interface on the gas phase side. H_a intermediates adsorbed on Pd membrane finally react with butadiene gas in gas-phase cell, offering the target products of butenes. The selective hydrogenation happens at the interface of Pd membrane to butadiene gas. Therefore, after careful and deep analysis of the reaction process, we did not find the “solvent effect” in this gas/solid interface heterogeneous reaction process.

Moreover, as for the proposition of reaction mechanism for electrochemistry-assisted hydrogenation in the submitted work, the strategy of “theoretical prediction followed by experiment verification” was employed. In detail, theoretically, DFT calculation was performed on the model derived from the XRD result (**Figure R6**), it shows that hydrogen atoms (H_a) prefer to stay at the interstitial void site of Pd surface, rather than at the top site of Pd, owing to their lower adsorption energy (E_{ads}). The E_{ads} of H_a at the interstitial void site was found to be higher than that of butadiene but lower than that of butenes, indicating an adsorption strength follows butadiene > H_a at the interstitial void site > butenes.

Moreover, the reactivity of different types of adsorbed hydrogens (i.e., the interstitial void site and the top site of Pd) for butadiene were fully evaluated via the Eley-Rideal (E-R) model and Langmuir-Hinshelwood (L-H) model, respectively. It shows that, as for Pd surface covered by H_a staying at the interstitial void site, butadiene has to be primarily adsorbed on Pd then further hydrogenated to butenes. Elimination of the intermediate butenes adsorption plays key role on deciding alkenes selectivity. Therefore, the H_a generated from water electrolysis, followed by

penetration through Pd and stays at the interstitial void site of Pd surface show an importance for improving the catalytic selectivity to alkenes.

Experimentally, XRD analysis on the Pd membrane after catalytic test shows that H_a stays at the interstitial void site of Pd, resulting in an expanded crystal structure of Pd (i.e., $PdH_{0.75}$ phase). Moreover, using butadiene as the feeding reactant, butadiene can be immediately converted to butenes at current density of water electrolysis as low as below 2 mA cm^{-2} . The spontaneous reaction over Pd surface after H_a penetration reveals that butadiene hydrogenation follows the Langmuir-Hinshelwood (L-H) model (i.e., reaction happens between two adsorbed species) rather than the Eley-Rideal (E-R) model (i.e., reaction happens between adsorbed species and molecule), which is an energy unfavorable for the reaction between H_a at the interstitial void site and butadiene calculated by the DFT method. Furthermore, difference from the high catalytic performance by feeding butadiene, we performed a supplementary experiment using alkenes (i.e., 2-butene and propene) as the feeding reactant in **Figure R7**, which shows that no alkenes hydrogenation to alkanes occurs. The inert activity for butene hydrogenation indicates that Pd surface is mainly covered by H_a at the interstitial void site as these species can prevent butene adsorption for further hydrogenation on Pd surface owing to their low E_{ads} . Therefore, the butane product in outlet gas of reactor is mainly formed from the over-hydrogenation (i.e., the L-H model) of adsorbed butenes, which are transformed from adsorbed butadiene molecule on Pd surface. As a result, it allows us propose the L-H mechanism for butadiene semi-hydrogenation reaction.

The highly comparable between the DFT prediction and the experimental result could allow us to properly propose a reasonable and reliable reaction mechanism. We hope our answers can satisfy you.

Figure R7. The digital curve of GC data during catalytic reaction with various alkenes feedstocks under different current densities at the GHSV of $\sim 5308 \text{ h}^{-1}$. (a) 0.6% of 2-butene in helium; (b) 3% of propene in helium;

4. *The authors have revealed the reaction mechanisms of butadiene hydrogenation by DFT calculations. Can the DFT calculations results be comparable with the experimental measurements?*

Answer: Thank you for your very helpful suggestion.

Following your suggestion, we have carefully checked that the DFT calculations results and our experimental results are highly comparable and are in excellent agreement.

Firstly, as for the hydrogen diffusion behavior in Pd membrane, two types of hydrogen atom penetration have been predicted by the DFT calculation, including a layer-by-layer penetration under slow hydrogen atom formation and a vertical penetration under rapid hydrogen atom formation in **Supplementary Fig. 6** and **Supplementary Fig. 16**. In experiment, the hydrogen atom electrochemical titration (HAET) test under different current densities for water electrolysis in **Supplementary Fig. 14** shows that, at j_1 below 0.15mA cm^{-2} (i.e., slow H_a generation), only one type of H_a penetration (with long time for H_a crossing over Pd membrane, t_b in **Supplementary Fig. 14d**) was recognized in the **Supplementary Fig. 15**. However, at j_1 above 0.15mA cm^{-2} (relative rapid H_a generation), two types of H_a penetration (with short time for H_a crossing over Pd membrane, t_b in **Supplementary Fig. 14d**) were recognized in the **Supplementary Fig. 15**. Therefore, these experimental results show the two different H_a penetration behaviors, which are in good agreement with the DFT predictions in **Supplementary Fig. 6** and **Supplementary Fig. 16**.

Moreover, DFT predicts that H_a prefers to stay at the interstitial void of Pd crystal owing to the relatively low adsorption energy (E_{ads} , **Fig. 4a**). It was verified by the XRD analysis, which shows a transformation in the crystal structure of palladium to palladium hydride ($\text{PdH}_{0.75}$), with an expanded crystal lattice by the existed H_a in the interstitial void of Pd crystal after catalytic test from XRD.

After H_a penetration to the Pd membrane interface on the gas-phase side, DFT calculation was used to evaluate the energy evolution of butadiene semi-hydrogenation to butene via different models, including the Langmuir-Hinshelwood (L-H) model (i.e., reaction happens between two adsorbed species) and the Eley-Rideal (E-R) model (i.e., reaction happens between adsorbed species and molecule). It shows that the E-R model is suitable for Pd surface with H_a at the top site of Pd, while the L-H model is energy favorable for Pd surface with H_a at the interstitial void of Pd. Experimentally, using butadiene as the feeding reactant, butadiene can be immediately converted to butenes at current density of water electrolysis as low as below 2 mA cm^{-2} . The spontaneous reaction over Pd surface after H_a penetration reveals that butadiene hydrogenation follows the L-H model.

In addition, the DFT calculation predicts that the H_a prefers to react with butadiene, rather than combines to H_2 . A supplementary experiment was performed to analyze the composition of outlet gas in **Figure R8**, which shows that almost no H_2 in the outlet gas. It agrees the DFT calculation.

Figure R8. The digital curve of GC-MS data during catalytic reaction. (a) scheme of analysis; (b) FID analysis; (c) TCD analysis; (d) MS analysis;

Last but not the least, DFT calculation shows that the adsorption strength (E_{ads}) of H_a at the interstitial void of Pd is higher than that of butadiene but lower than that of butenes. H_a can prevent the adsorption of butenes for further reaction. This was verified by a supplementary experiment. Different from using butadiene, the supplementary experiment uses butene as reactant (**Figure R7a**). However, it shows no catalytic performance. It indicates that the formed butane in the outlet gas is mainly from the over-hydrogenation of adsorbed butene through the L-H model.

Therefore, the DFT calculations results and the experimental results are in excellent agreement. We hope our answers can convince you.

Finally, we would like to thank this reviewer for his/her precious time and professionalism devoted to the evaluation of our manuscript. His/her valuable and constructive comments indeed largely improve the quality of our manuscript.

REVIEWERS' COMMENTS

Reviewer #1 (Remarks to the Author):

Revisions were made properly. I feel the present form is suitable for publication in Nature Communications.

Reviewer #4 (Remarks to the Author):

Overall, this is a well-conceived study. The concept aligns well with current trends in utilizing clean renewable energy sources for chemical productions. Therefore, this work shall be well received by the community. The manuscript is concise and the results can support the authors' conclusion.

I have read the revised manuscript and the authors' response to the previous round of review. I don't have too many additional comments. My main question that does not seem to be addressed is if the Pd is not single crystal, would it still be appropriate to perform DFT calculations on Pd(111) surface. Perhaps the conclusions may not change. For the sake of completeness, the authors may include step and kink surface geometries based on Pd(211) or Pd(335) facets.

Also, there are still typos in the manuscript. For example, in line 145 on page 8, 'detailed discussed' should be 'detailed discussion'. The authors should use relevant programs or profession editing service to eliminate such problems.

Responses to the comments from Reviewers 1 and 4

Reviewer #1 (Remarks to the Author):

Revisions were made properly. I feel the present form is suitable for publication in Nature Communications.

Answer: Thank you very much for your very positive comment.

We are very grateful to you for your very positive and encouraging comment. With your encouragement, we will continue to work hard to provide more innovant research works.

We would like to thank this reviewer for his/her precious time and professionalism devoted to the evaluation of our manuscript to give very positive comment to our manuscript.

Reviewer #4 (Remarks to the Author):

Overall, this is a well-conceived study. The concept aligns well with current trends in utilizing clean renewable energy sources for chemical productions. Therefore, this work shall be well received by the community. The manuscript is concise and the results can support the authors' conclusion.

- 1) I have read the revised manuscript and the authors' response to the previous round of review. I don't have too many additional comments. My main question that does not seem to be addressed is if the Pd is not single crystal, would it still be appropriate to perform DFT calculations on Pd(111) surface. Perhaps the conclusions may not change. For the sake of completeness, the authors may include step and kink surface geometries based on Pd(211) or Pd(335) facets.
- 2) Also, there are still typos in the manuscript. For example, in line 145 on page 8, 'detailed discussed' should be 'detailed discussion'. The authors should use relevant programs or profession editing service to eliminate such problems

Answer: We thank you for your positive comments and your constructive inputs and suggestions, which are very helpful in improving the quality of our manuscript. We have carefully and thoroughly revised the manuscript by taking each of your suggestions and comments into account, including carrying out supplementary DFT calculations to improve the quality and completeness of our manuscript. Moreover, the English writing in this revised manuscript was carefully checked by our English colleague. We are sure that we have eliminated remaining typos and the clarity of manuscript has been further improved. We do hope that our supplementary DFT calculations, explanations and thorough revisions based on

your questions and your professional advices will satisfy you.

Please find below our point-by-point answers to your comments and suggestions. *Your comments and questions are written in blue italics* and our responses in “black”. For your facility, we have 1) shown at the end of our answer to each of your questions the changes we have made in the revised manuscript and supplementary information with page number and the numbers of lines and 2) also highlighted **in red colour** all the changes we have made in the revised manuscript.

General comment:

Overall, this is a well-conceived study. The concept aligns well with current trends in utilizing clean renewable energy sources for chemical productions. Therefore, this work shall be well received by the community. The manuscript is concise and the results can support the authors' conclusion.

Answer: Thank you very much for your positive comment. We have modified the manuscript to take very carefully your specific comments into account.

Specific comments:

1. I have read the revised manuscript and the authors' response to the previous round of review. I don't have too many additional comments. My main question that does not seem to be addressed is if the Pd is not single crystal, would it still be appropriate to perform DFT calculations on Pd(111) surface. Perhaps the conclusions may not change. For the sake of completeness, the authors may include step and kink surface geometries based on Pd(211) or Pd(335) facets.

Answer: Thank you for your very helpful suggestion.

Following your valuable suggestion, additional DFT calculations were performed on Pd(335) surface, including the step and kink surface geometries during butadiene hydrogenation (**Figure R9**). The DFT results agree well with your professional opinion that it does not change the conclusion.

Figure R9. Activation energy of butadiene hydrogenation over Pd(335), with corresponded structures of transition state (TS);

In details, a slab of Pd(335), containing six atomic layers with a (3×1) unit cell and separated by 15 Å vacuum space, was built and optimized by vienna ab-initio simulation package (VASP) based on density functional theory. The reaction process of butadiene hydrogenation, with 1-butene and butane as the probe products of semi-hydrogenation and full-hydrogenation, was simulated on the slab of Pd(335). It was found that the activation energy (E_a) of the four “half-reaction” in butadiene hydrogenation are 0.45, 0.80, 0.54 and 0.73eV, which are similar as those on Pd(111) in the **Supplementary Fig. 9(b)**.

Therefore, according to your professional suggestion, the DFT calculations have been made and added in the Supplementary as **Supplementary Fig. 9(d)**. Moreover, the corresponded descriptions have been added in the main manuscript (Line 176, Page.9), the experimental part (Line 509-511, Page 23). We hope our answers can satisfy you.

2. Also, there are still typos in the manuscript. For example, in line 145 on page 8, ‘detailed discussed’ should be ‘detailed discussion’. The authors should use relevant programs or profession editing service to eliminate such problems

Answer: Thank you for your very helpful suggestion.

Based on your great suggestion, we have asked our English colleagues to carefully checked the English writing in this revised manuscript. We hope the English writing in this manuscript could satisfy you.

Finally, we would like to thank this reviewer for his/her precious time and professionalism devoted to the evaluation of our manuscript. His/her valuable and constructive comments indeed largely improve the quality of our manuscript.